# Metallo-Supramolecular Complexation Behavior of Terpyridine- and Ferrocene-Based Polymers in Solution—A Molecular Hydrodynamics Perspective

**DOI:** 10.3390/polym14050944

**Published:** 2022-02-26

**Authors:** Igor Perevyazko, Nina Mikusheva, Alexey Lezov, Alexander Gubarev, Marcel Enke, Andreas Winter, Ulrich S. Schubert, Nikolay Tsvetkov

**Affiliations:** 1Department of Molecular Biophysics and Polymer Physics, St. Petersburg University, Universitetskaya Nab. 7/9, 199034 Saint-Petersburg, Russia; n.mikusheva@spbu.ru (N.M.); a.a.lezov@spbu.ru (A.L.); a.gubarev@spbu.ru (A.G.); 2Laboratory of Organic and Macromolecular Chemistry (IOMC), Friedrich Schiller University Jena, Humboldt Str. 10, 07743 Jena, Germany; marcel.enke@uni-jena.de (M.E.); andreas.winter@uni-jena.de (A.W.); 3Jena Center for Soft Matter (JCSM), Friedrich Schiller University Jena, Philosophenweg 7, 07743 Jena, Germany

**Keywords:** metallo-supramolecular complexes, molecular hydrodynamics, solution properties, analytical ultracentrifugation, conformation, metal-ligand interactions

## Abstract

The contribution deals with the synthesis of the poly(methacrylate)-based copolymers, which contain ferrocene and/or terpyridine moieties in the side chains, and the subsequent analysis of their self-assembly behavior upon supramolecular/coordination interactions with Eu^3+^ and Pd^2+^ ions in dilute solutions. Both metal ions provoke intra and inter molecular complexation that results in the formation of large supra-macromolecular assembles of different conformation/shapes. By applying complementary analytical approaches (i.e., sedimentation-diffusion analysis in the analytical ultracentrifuge, dynamic light scattering, viscosity and density measurements, morphology studies by electron microscopy), a map of possible conformational states/shapes was drawn and the corresponding fundamental hydrodynamic and macromolecular characteristics of metallo-supramolecular assemblies at various ligand-to-ion molar concentration ratios (M/L) in extremely dilute polymer solutions (c[η]≈0.006) were determined. It was shown that intramolecular complexation is already detected at (L≈0.1), while at M/L>0.5 solution/suspension precipitates. Extreme aggregation/agglomeration behavior of such dilute polymer solutions at relatively “high” metal ion content is explained from the perspective of polymer-solvent and charge interactions that will accompany the intramolecular complexation due to the coordination interactions.

## 1. Introduction

Metallopolymers represent a special class of macromolecules in which the intrinsic properties of conventional polymers (e.g., their mechanics and processability) are combined with those due to the presence of the metal centers (e.g., catalytic, magnetic, photophysical and electrochemical behavior) [1]. Within the last decades, a range of applications has been established on the basis of metal-containing polymers; these include inter alia opto-electronic devices for energy storage or interconversion, immobilized catalysts, pharmaceuticals, self-healing materials and precursor materials for nanoparticle preparation [1,2,3,4]. Research in these fields has mainly been focused on optimizing the preparation, enabling a reliable characterization and tuning the performance of the materials with respect to the fields of interest. An understanding of how the distribution of charged metal sites along a polymer chain affects the conformation and/or aggregation in solution is of central importance to optimize the structure and, thereby, to improve the property of interest (e.g., intra-/interchain energy-transfer processes, accessibility of catalytically or pharmaceutically active centers, etc.). However, the molecular properties of polymers containing metal ligands in the side chains and the knowledge on the corresponding mechanism of supramolecular assembling in dilute solution are far from being fully understood from the perspective of polymer physics and molecular hydrodynamics, since such polymer systems have hardly been looked at in a systematic fashion [5,6,7,8,9,10,11,12]. Consequently, the presented research deals with the investigation of the complexation behavior of tpy- and Fc-containing methacrylate-based copolymers in very dilute solution under the addition of Eu^3+^ and Pd^2+^ metal ions (tpy: 2,2′:6′,2″-terpyridine; Fc: ferrocene). The investigation was performed from the basic perspectives of polymer physics and molecular hydrodynamic applying mutually reinforcing set of classical and modern analytical techniques such as analytical ultracentrifugation, dynamic light scattering, viscosity and density measurements, scanning microscopy, etc.

## 2. Material and Methods

### 2.1. Materials

All chemicals and reagents were purchased from Sigma Aldrich (Taufkirchen, Germany), TCI (Eschborn, Germany) or Fisher Scientific (Schwerte, Germany). For monomer synthesis all compounds were employed without further purification. Prior to polymerization, toluene was purified by distillation over calcium hydride. The initiator, 2,2′-azobisisobutyronitrile (AIBN), was purified by recrystallization from ethanol. The glassware used for the syntheses was oven dried at 110 °C. The tpy- and Fc-containing monomers **1** and **2**, respectively, were prepared according to procedures published elsewhere [13,14].

### 2.2. General Protocol for the Polymer Synthesis via RAFT

A microwave vial (5 mL) was charged with the respective monomers and a toluene stock solution, which contained AIBN (the initiator) and CPDB (the chain-transfer agent). The resulting solution was degassed by purging with a N_2_ stream for 30 min. Subsequently, the sealed vial was heated to 70 °C for 12 h. The crude polymer was precipitated by dropping the cooled reaction mixture into MeOH (20 mL). The monomer was removed by dialysis in THF using a size-exclusion membrane (MWCO 3300 Da).

### 2.3. Structural Characterization of the Polymers

^1^H-NMR spectra were recorded on an AVANCE I 300 MHz instrument (Bruker, Ettlingen, Germany) in CD_2_Cl_2_ (euriso top) at $T=25{\;}^{\circ }C$. Chemical shifts are reported in ppm and are referenced using the residual solvent signal. Size-exclusion chromatography (SEC) was carried out using a 20er series instrument (Shimadzu, Jena, Germany) comprising of a DGU-14A degasser, a CBM-20A controller, a LC-20AD pump, a SIL-20AHT auto sampler and a CTO-10AC vp oven. For the separation, a PSS SDV guard column (1000 Å to 100,000 Å; 5 µm particle size) was used. The signal detection occurred via a RID-10A detector. The samples were analyzed in a chloroform/isopropanol/triethylamine solvent mixture (94:2:4 ratio) at $T=40{\;}^{\circ }C$ and a flow rate of 1 mL/min. The molar mass and dispersity were determined using a linear PMMA calibration (400 to $10^{6}$ ${\mathrm{gmol}}^{-1}$, PSS GmbH, Mainz, Germany).

### 2.4. Analytical Ultracentrifugation (AUC) 

Sedimentation velocity experiments were performed with a ProteomeLab XLI AUC system (Beckman Coulter, Brea, CA, USA), using double-sector Epon or Aluminum centerpieces with a 12 mm optical solution path length and a four-hole rotor (An-60Ti). The rotor speed was varied between 1500 to 40,000 rpm, depending on the sample. The sector-shaped cell compartments were filled with ~420 μL of sample solution and ~440 μL of the solvent (THF) in the reference sector. Before the run, the rotor was equilibrated for approximately 1 h at $T=20.0{\;}^{\circ }C$ in the AUC. Sedimentation velocity profiles were recorded by the absorbance and interference/refractive index (RI). 

### 2.5. Dynamic- and Static-Light Scattering

Batch dynamic light scattering (DLS) and static light scattering (SLS) experiments were carried out using a PhotoCor-Complex apparatus (Photocor Instruments Inc., Moscow, Russia), equipped with a real-time correlator (288 channels, minimal correlation time of τ=10 ns). The wavelength of the laser was λ=405 nm; the intensity fluctuations were recorded at scattering angles (ϑ) ranging from a $30^{{}^{\circ}}$ to a $140^{{}^{\circ}}$ scattering angle. The experiments were performed at a temperature of $T=20.0{\;}^{\circ }C$. The obtained autocorrelation functions of the scattered light intensities were processed using the DynaLS Software (Photocor, Moscow, Russia), which provides the respective distributions ρ(τ) of the scattered light intensities by the respective known relaxation times, τ. The dependence between 1/τ (where *τ* is the maximum intensity of the ρ(τ) distribution) and the squared scattering vector $q^{2}={\left(4\pi n/\lambda \;\sin\left(\vartheta /2\right)\right)}^{2}$ for all the studied macromolecule populations was observed being a straight line passing through the plot origin, i.e., representing the translational diffusional processes recorded. The translational diffusion coefficients at the particularly measured macromolecule concentrations in solutions, D, was calculated from the slope of this recorded line according to the following relationship: $1/\tau =Dq^{2}$ [15].

### 2.6. Densitometry

The density measurements for partial specific volume determination were carried out in a density meter (DMA 5000M, Anton Paar, Graz, Austria) according to the classical procedure of Kratky et al. [16] that is commonly applied in diverse recent experimental studies [17,18,19,20,21]. 

## 3. Results and Discussion

The polymers containing tpy and/or Fc moieties in the side chains, were prepared via a reversible addition-fragmentation chain-transfer (RAFT) polymerization. This particular polymerization method has proven its versatility for the synthesis of defined polymers, including metal-containing ones [1,22]. For example, tpy-equipped homopolymers and copolymers were prepared via RAFT processes [23,24]. The same holds true for Fc-containing polymers, which can inter-alia be prepared in a highly controlled fashion by RAFT [25]. Due to the high compatibility of the RAFT polymerization with tpy and Fc moieties, we opted for this approach and the polymers TerPol and TerFerCop were prepared by standard RAFT conditions using AIBN and CPDB, as the initiator and chain-transfer agent, respectively (CPDB: 2-cyano-2-propylbenzodithioate; Figure 1).

The structural characterization of TerPol and TerFerCop was performed by ^1^H-NMR spectroscopy and size-exclusion chromatography (SEC). The latter revealed relatively narrow dispersities (Ð of ca. 1.4) as well as molar masses (M_n_) which were in fair agreement with the applied monomer-to-initiator (M/I) ratios. The SEC curves, as recorded with a refractive-index (RI) detector, are depicted in Figure 1. According to a linear PMMA calibration, the $M_{n}$ values of the polymers were $39,000\;{\mathrm{gmol}}^{-1}$ and $29,000\;g/{\mathrm{mol}}^{-1}$ for TerPol and TerFerCop, respectively. The ^1^H-NMR spectra (Figure 2, Appendix A) clearly revealed the incorporation of the tpy and the Fc moieties into the polymer. In the case of TerFerCop, the tpy-to-Fc ratio of 1:1.24 was calculated from the integrals of the characteristic tpy and Fc signals in the range of 7 to 9 ppm and 4.75 ppm, respectively (see the Appendix A).

From the perspective of classical polymer physics/molecular hydrodynamics, in order to understand fundamental structure-property correlations that determine the formation and behavior of the polymer-based metallo-supramolecular assemblies at different metal to ligand ratio (M/L) in solution, systematic investigations/determinations of certain solution characteristics that are related to the basic macromolecular parameters (molar mass (M), size (R), shape) will be required. Such fundamental solution characteristics Equation (1) will basically include the sedimentation coefficient–s, S, the translation diffusion coefficient $D,\;\left[{\mathrm{cm}}^{2}s^{-1}\right]$ and the intrinsic viscosity $\left[\eta \right],\;\left[{\mathrm{cm}}^{3}g^{-1}\right]$ [26,27]:(1)$$s=\frac{M\left(1-\upsilon \rho_{0}\right)}{N_{A}6\pi \eta_{0}R}D=\frac{k_{B}T}{6\pi \eta_{0}R}\left[\eta \right]=\Phi \frac{R_{\eta }^{3}}{M}$$
where M—the molar mass, R—the hydrodynamic radius, υ—the partial specific volume, $\rho_{0}$—solvent density, $\eta_{0}$—dynamic viscosity of the solvent, $N_{A}$—Avogadro’s number, $k_{B}$—Boltzmann constant, Φ—Flory hydrodynamic parameter, $R_{\eta }$—viscosity hydrodynamic size. 

Due to the basic inability to experimentally determine intrinsic viscosities in case of very dilute solutions/suspensions in the present study, we will mainly focus the attention on the sedimentation-diffusion analysis and associated with it measurements (i.e., partial specific volume), while the intrinsic viscosity measurements were determined only for the tpy-based homopolymer and its copolymer with additional Fc units in order to estimate the degree of dilution (the Debay parameter c[η]). At first, we will briefly look on the characteristics of the initial polymers (TerPol and TerFerCop), their molecular parameters/distributions will serve as reference points for later comparison/discussions. Furthermore, in the first part we will as well introduce main approaches used for the analysis of the sedimentation velocity data obtained by the analytical ultracentrifugation. After that we move on to the results and discussion of the complexation studies of the tpy- containing polymers with Eu^3+^ and Pd^2+^ salts ((Eu(CF_3_SO_3_)_3_ and PdCl_2_ (CH_3_CN)_2_).

### 3.1. Initial Polymers and Sedimentation Data Analysis

In Figure 3 distributions (**A**,**B**) of the sedimentation coefficients for the TerPol and TerFerCop are accompanied by the sets of the recorded sedimentation profiles (concentration signal over radial distance) (**C**,**D**) that correspond to the lowest analyzed solute concentration (c≈0.03%).

The sedimentation data was analyzed using the Sedfit Software [28]. The core problem in the analysis of the sedimentation velocity data, sine the AUC and the corresponding physical/theoretical background was established [29,30,31] lies within the solutions of the fundamental Lamm equation Equation (2):(2)$$\frac{\partial c}{\partial t}=\frac{1}{r}\frac{\partial }{\partial r}\left[\left(D\frac{\partial c}{\partial r}-\omega^{2}rsc\right)r\right]$$
where c is the solute concentration, t is the time, r is the radial distance, $\omega^{2}={\left(2\pi n/60\right)}^{2}$ is the angular velocity, D is the translation diffusion coefficient, s is the sedimentation coefficient. 

This equation describes the evolution of the sedimenting species in time (their respective concentration profiles) during the centrifugation process, and it contains two very important physical characteristics–the sedimentation coefficient, s, and the translation diffusion coefficient D that describe corresponding physical processes in the solute. Since the Lamm equation is a partial differential equation it can be numerically solved, providing thereby information on both the sedimentation and the diffusion [32,33,34]. The initial information that is analyzed is the set of the sedimentation profiles, representing the radial position of the sedimenting particles/molecules in time. In Figure 3, the latter ones are shown in graphs (**C**,**D**), while graphs (**A**,**B**) display the corresponding distributions obtained by different analytical approaches implemented into Sedfit—the c(s) [35], the $c\left(s,f/f_{0}\right)$ [36] and $ls-g^{*}\left(s\right)$ [37] analysis. The c(s) and $c\left(s,f/f_{0}\right)$ models numerically solve the Lamm equation, while the $ls-g^{*}\left(s\right)$ analysis does not, considering species that experience no diffusion, and the data set (sedimentation profiles) is analyzed basically using the least square boundary approach [38]. Furthermore, it should also be noted that the c(s) model is intrinsically confined within the spherical approach ($s\sim M^{2/3}$), providing also only average diffusion coefficient (or $f/f_{0}$) over all spices presented in solution, while the $c\left(s,f/f_{0}\right)$ is known to be a model free approach having no any presumptions on the particle shapes that as well provides corresponding distribution of the diffusion coefficients (or $f/f_{0}$). 

Coming back to Figure 3, it can be seen that in case of c(s) and $c\left(s,f/f_{0}\right)$ models, the TerPol and TerFerCop are characterized by multimodal distributions. Such discretization of the sedimentation distributions is a result of the performed Lamm equation analysis. The presence of discreet number of individual peaks in the distributions of such synthetic polymers does not necessarily assume the physical presence of such distinct molecular fractions in the solution, since the statistical nature of the polymerization process does not favor the synthesis of such individual fractions, but rather a continuous distribution of molecular species is expected. The multimodality and overall broadness of the distributions should be considered in the first place as a representation of relatively high molecular dispersity of the studied polymer solutions. The actual distribution may be represented as a single broad peak that covers entire range of the sedimentation coefficients. Such distributions can be obtained by the $ls-g^{*}\left(s\right)$ analysis. Its application is favorable for systems with relatively low diffusion spreading of the sedimentation boundary, i.e., high sedimentation rate, however in spite of the more natural distribution obtained by the $ls-g^{*}\left(s\right)$ we will still stick to the higher resolution distributions obtained by c(s) and $c\left(s,f/f_{0}\right)$ models in the further analysis, since this analysis as well provide information about the diffusion coefficients/frictional ratios. The average sedimentation coefficients obtained by the $ls-g^{*}\left(s\right)$, c(s) or $c\left(s,f/f_{0}\right)$ will be the same within the experimental error.

The diffusion coefficients, in case of c(s) and $c\left(s,f/f_{0}\right)$, are initially “represented” in the form of the frictional ratios ($f/f_{0}$)—very popular construct in the molecular biophysics, where $f=6\pi \eta_{o}R$—is the translation friction coefficient of the particle/molecule under the study and $f_{0}=6\pi \eta_{o}R_{sph}$ is that of the spherical particle having the same mass and the density as the studied one. Such a construct initially is used to estimate the overall asymmetry of studied objects based on some model systems, for example prolate or oblate ellipsoid of revolution or rod/cylinder [39,40,41]. The diffusion coefficient D can then be reestablished as:(3)$$D_{0}=\frac{k_{B}T{\left(1-\upsilon \rho \right)}^{1/2}}{\eta_{0}^{3/2}9\pi \sqrt{2}{\left(f/f_{0}\right)}^{3/2}{\left(s_{0}\upsilon \right)}^{1/2}}$$

The estimated sedimentation and diffusion coefficients together with the determined partial specific volumes are then can be used to calculated the molar masses via the Svedberg equation Equation (4) (Table 1). The partial specific volume υ of the initial polymers was determined by the classical density measurements of differently concentrated polymer solutions [16,27] and constitutes $0.75\;{\mathrm{cm}}^{3}g^{-1}$ and $0.65{\;\mathrm{cm}}^{3}g^{-1}$ for the TerPol and TerFerCop, respectively.
(4)$$M=\frac{RTs_{0}}{\left(1-\upsilon \rho_{0}\right)D_{0}}=9\pi \sqrt{2}N_{A}{\left(\left[s\right]\;f/f_{0}\right)}^{3/2}\sqrt{\upsilon }$$
where $\left[s\right]=\frac{s_{0}\eta_{0}}{\left(1-\upsilon \rho_{0}\right)}$—intrinsic sedimentation coefficient, $f/f_{0}$—frictional ratio, υ—partial specific volume, $\rho_{0}$—density of a solvent, $\eta_{0}$—dynamic viscosity of the solvent, $N_{A}$—Avogadro’s number.

The found molar masses of 38,000 ± 3000 gmol^−1^ and 70,000 ± 10,000 gmol^−1^ for the TerPol and TerFerCop systems by the sedimentation-diffusion analysis on the first site are markedly different from the ones obtained by the conventional SEC analysis. But taking into account rather high dispersity of the polymers and the use of linear PMMA samples as standards for the SEC analysis we may conclude on the reasonable agreement between the values. For the further analysis we will use the values obtained by the sedimentation-diffusion analysis ($M_{sD}$), since it is considered to be an “absolute” characterization technique.

### 3.2. Complexation Studies in Solution

The complexes were prepared by mixing equal volumes of corresponding polymer and salt solutions ((Eu(CF_3_SO_3_)_3_ and PdCl_2_ (CH_3_CN)_2_) at appropriate concentrations that cover the following range of metal to ligand molar ratios: 0.01≤M/L≤0.5. In order to promote formation of individual intra-molecular complexes rather than intermolecular network like structures and to reduce possible non-ideal effects at high polymer concentrations, the complexation was studied in a very dilute polymer solutions were the corresponding Debay parameter—c[η] [27], that characterizes the volume fraction occupied by the macromolecules in solution, was ~0.006, meaning that only ~0.6% of the total solution volume is occupied by the macromolecules. In our further discussion we will first look on the upper half of the M/L range (0.1≤M/L≤0.5) that correspond to the molar salt concentrations of $\sim 1\times 10^{-4}\;M$ to $\sim 5\times 10^{-4}\;M$. It should be noted that for all studied polymer-metal systems, no stable solutions/suspensions were obtained at M/L>0.5. After that we will focus on the extra small salt contents (0.01<M/L<0.1) and subsequently summarize the data for further analysis and discussion.

Figure 4 shows evolution of the distributions of the sedimentation coefficients for the TerPol and TerFerCop systems coordinated with Eu^3+^ and Pd^2+^ ions at the M/L ratios of approx. 0.1, 0.3 and 0.5. It can be seen that in general all studied systems follow the same trend revealing a shift towards higher sedimentation coefficients with increasing salt content. At the M/L ratios ~0.1, the distributions are still close to the range of the sedimentation coefficients found for the initial macromolecules (1≤s,S≤40), with yet clear changes in the composition and position of the main sedimenting species. At this stage we believe, we still have a molecular like solutions were the metallo-polymer complexes are still can be considered to be in dissolved state, while once we increase the salt content approximately twice, the substantial increase of the average sedimentation coefficients $100\leq s_{av},S\leq 1000$ shifts us towards suspension like systems. However, the distributions, in all cases, clearly show the presence of relatively low molar mass fractions, at the same time the overall dispersity of the solution/suspensions is much increased. Further increase of the salt content close to M/L≈0.4−0.5 results in the formation of large supramolecular assembles/complexes having extremely broad distributions with the average sedimentation coefficients *s* >> 1000 *S*. 

At high salt concentrations we observe full “transformation” from the molecular like solutions were the macromolecules are considered to be dissolved towards colloidal like system that characterizes by one or another type of phase segregation in solution/suspension, such as for example polymer-based nano/macro particles [42]. Also, we do not see any specific changes in the overall picture considering different polymer systems/metal ions. It is only notable that in general the TerFerCop in comparison to TerPol systems revealing generally higher values of the sedimentation coefficients. It is also notable that in comparison to Eu^3+^, the Pd^2+^ systems show as well generally higher values of the sedimentation coefficients and overall dispersity of the distribution. This is quite interesting, since while the Eu^3+^ ions are capable to coordinate several tpy groups, the Pd^2+^ ions are known to form complexes with tpy of square-planar geometry and in the first approximation, since the coordination number of Pd^2+^ is only 4 and it should not be able to coordinate several tpy ligands. Further discussion regarding the complexation mechanism will be given later (vide infra). At almost an order of magnitude lower the salt content more gradual changes/evolution of the initial distributions which are shown in the Figure 5 and Appendix A were observed. 

It can be seen that at the lowest salt content (M/L≈0.02), the corresponding distributions has either clearly shifted keeping the composition the same, like in the case of terpyridine homopolymer system with Eu^3+^ ions, or became notably broader with a shoulder towards higher molar mass range (higher sedimentation coefficients) like in the case of TerFerCop Eu^3+^ complexes. The majority of the material (~90%) in the distributions (Figure 5) can surely be attributed as dissolved/solvated individual supra-molecular coils. At the same time, we yet observe some fractions of extra higher molar mass aggregates (<10%), that likely represent already formed/aggregated supramolecular complexes of high molar mass. Further analysis, in particular determination of the average molar masses and/or hydrodynamic sizes/shapes of the supramolecular complexes will require the knowledge of their partial specific volume or density of the sedimenting objects.

### 3.3. Partial Specific Volume 

Fundamentally, the partial specific volume (υ, p.s.v.) describes a volume change of a system upon an addition of a mass unity, at constant temperature (T), pressure (P) and mass of other constituents j:(5)$$\upsilon_{i}={\left(\frac{\partial V}{\partial g_{i}}\right)}_{T,P,j}\left(i\neq j\right)$$

In the first approximation it is often considered as the value reciprocal to the density (υ=1ρ), with the corresponding units of $\left[{\mathrm{cm}}^{3}g^{-1}\right]$. For the homologous series of polymers, one should expect an independency of the p.s.v. from the molar mass, but the value may be influenced by the solvent and charge interactions, solvation rate etc. [27]. 

The initial polymers, as was as mentioned above have the values of $\upsilon =0.75\;{\mathrm{cm}}^{3}g^{-1}$ for TerPol and $\upsilon =0.65\;{\mathrm{cm}}^{3}g^{-1}$ for TerFerCop, that were determined by the classical density measurements. We see that the presence of the ferrocene group in the polymer chain results in the notable decrease of the υ in comparison to terpyridine homopolymer, which is basically due to the presence of “heavy” metal ion.

Partial specific volume of the supramolecular assembles at different M/L ratios were determined by the density variation/contrast approach [43], since the classical density approach can hardly be applied for such systems. The density variation approach comes down to the sedimentation velocity experiments in isotopically different solvents (THF and $THF_{d8}$ in the current case). The partial specific volume can then be calculated). Assuming the same polymer-solvent interactions, the recorded difference in the sedimentation coefficients, due to the higher viscosity and density of the deuterated solvents, the p.s.v. can be calculated as Equation (6):(6)$$\upsilon =\frac{s_{2}\eta_{2}-s_{1}\eta_{1}}{s_{2}\eta_{2}\rho_{1}-s_{1}\eta_{1}\rho_{2}}$$
where $s_{1}$ and $s_{2}$ are the sedimentation coefficients in THF and $THF_{d8}$ and $\eta_{1}=0.499\;\mathrm{cP}$, $\rho_{1}=0.8876{\;\mathrm{gcm}}^{-3}$ and $\eta_{2}=0.519\;\mathrm{cP}\;\left(50\%\;THF_{d8}\right),\;0.528\;\mathrm{cP}\left(75\%\;THF_{d8}\right)$, $\rho_{2}=0.93898{\;\mathrm{gcm}}^{-3}\;\left(50\%\;THF_{d8}\right),\;0.96554{\;\mathrm{gcm}}^{-3}\left(75\%\;THF_{d8}\right)$ are the dynamic viscosities and densities of the used THF and THF-d_8_, respectively. The estimated values are shown in Figure 6 as a function of metal salt concentration.

It is notable that already at the very low metal-salt concentrations the υ has slightly decreased, which is reasonable, due to the appearance of the heavy-metal ions on the polymeric chain. However, at the highest salt concentration (M/L=0.4−0.5) we obsreve a notable increase of υ to values of $0.87\;{\mathrm{cm}}^{3}g^{-1}$ for TerPol and to $0.76{\;\mathrm{cm}}^{3}g^{-1}$ for TerFerCop, signifying that the overall density of the supramolecular assembles, in spite of the presence of the metal ions has decreased. Having at hand experimentally determined values of υ allow us to determine the corresponding hydrodynamic sizes, diffusion coefficients, molar masses and overall asymmetry of the supramolecular complexes (vide infra). But at first, we will summarize the results of the sedimentation velocity experiments in terms of the dependences of the average sedimentation coefficients and estimated average frictional ratios $f/f_{sph}$ as a function of M/L ratio (Figure 7).

Both graphs are plotted in the double-logarithmic manner. It is seen that the sedimentation dependence generally resembles the overall picture shown by the corresponding distributions discussed above—gradual increase of the average sedimentation coefficients upon increase of the metal content, which can be well fitted by the second order polynomial function. A substantial increase of the average sedimentation coefficients begins at M/L>0.1. The frictional ratio dependence, in its way, generally resemble/supports the characteristic behavior of the sedimentation coefficients, showing small, yet notable decrease of the average $f/f_{0}$ values at M/L>0.1, that could be a reflection of an active intermolecular complexation/agglomeration of the initially formed individual supramolecular assembles. Furthermore, it is also seen that in comparison to the initial polymers, the average $f/f_{0}$ in the first half of the M/L range (from ~0.02 to ~0.1) showing as well notable decrease. This may reflect an initial compactization of the polymer coils due to the intramolecular compactization caused by coordination of one metal ion with multiple ligand groups which tends macromolecule to reduce its volume [5]. However, despite the overall decrease of the $f/f_{0}$ ratio, the average values tend to stay at the end at 1.5±0.1, which may presume the presence of rather asymmetrical structures. It can then be for example transformed into the axial ratio (p=a/b) of the corresponding prolate ellipsoid of revolution. which in our case (assuming no solvation) will be a/b (prolate)≈9, which in general fits well with corresponding structures observed by the electron microscopy (Figure 8 and Appendix A) showing the presence of variously assembled complexes/structures. Specifically, in Figure 8, the SEM images visualize the TerFerCop-Eu^3+^ complexes at M/L ratios of ~0.1, ~0.3 and ~0.5. The SEM images are accompanied by the corresponding high-resolution distributions of the $f/f_{0}$ over the sedimentation coefficients obtained by the $c\left(s,f/f_{0}\right)$ analysis. It is seen that at (M/L=0.1), we have predominantly individual particles with an average diameter of 15±10 nm. At M/L=0.3, a large number of assembled/aggregated supramolecular particles with the sizes around 50 nm and predominantly worm/rod-like shapes. Separate structures consist of ~2 to ~10 individual particles.

The corresponding frictional ratio distribution does show, that while the majority of the material has $f/f_{0}\leq 1.3$, there are some distinct fractions having higher than average $f/f_{0}$ values (up to ~1.8), that can be associated with the presence of worm/rod like structures that we observe on the surface in the solution/suspension as well. At the M/L=0.5 further aggregation and formation of long wormlike and other type of aggregated/assembled structures/surfaces consisting of individual particles with the sizes ~150 nm was observed (Appendix A). Sedimentation velocity experiments of the metal-polymer suspensions at different periods of time, shows that the system is very dynamic. Figure 9 demonstrates the comparison of the sedimentation and $f/f_{0}$ distributions at the M/L~0.5 for the TerFerCop-Eu^3+^ system. We observe that substantial increase of the average sedimentation coefficients is accompanied by the corresponding increase of the frictional ratios, signifying formation of larger, but at the same time more asymmetrical structures/complexes.

Further discussions regarding complexation behavior will be given referring to Figure 10 which shows number of macromolecules forming one supramolecular complex/aggregate and the average hydrodynamic diameter as a function of M/L ratio, in a double logarithmic manner. 

The number of macromolecules was calculated based on the molar mass estimations of the complexes via the Svedberg equation Equation (4) using the average sedimentation and diffusion coefficients (or $f/f_{0}$) found for the complexes and comparing it to the molar masses of the initial polymers. The hydrodynamic sizes, on the one side were calculated from the sedimentation velocity data as Equation (7):(7)$$d=3\sqrt{2}\sqrt{\left[s\right]\upsilon }{\left(f/f_{0}\right)}^{3/2}$$
where $\left[s\right]=\frac{s\eta_{0}}{\left(1-\upsilon \rho_{0}\right)}$ is the intrinsic sedimentation coefficient, $\eta_{0}$, $\rho_{0}$ are the dynamic viscosity and density of the solvent, υ is the partial specific volume.

The presence of the term $f/f_{0}$ in Equation (7) does allow one, to a certain extent, to account for the asymmetry of the complexes, collapsing to spherical limit at $f/f_{0}=1$. For the simplicity Figure 10 shows the average hydrodynamic diameters calculated by Equation (7), however the sizes of certain individual fractions can as well be obtained. On the other side the hydrodynamic sizes were monitored/determined by the batch DLS measurements, the results are summarized in Appendix A and generally agrees well with the sedimentation data showing presence of two main fractions—the large one and the small one, that can generally be attributed to the “individual” or molecular supramolecular assembles and their large macromolecular aggregates. Moreover, due to the higher sensitivity for larger sizes, the DLS analysis shows that the large supramolecular aggregates (~200 nm) are already present at M/L≈0.02. 

The dependence (Figure 10) was divided into the several zones—A, B, C, D and F. The first “green” zone A describes the area of extremely low salt content (0.02≤M/L<0.1). The average molar mass of the complexes in this zone generally does not exceed the double amount of the initial polymer’s molar masses. Thus, we may assume that at this range of M/L ratios, predominantly, intramolecular type of complexation occurs that favors formation of unimolecular supramolecular assembles within one polymer chain or more precisely within one polymer coil. At the same time, due to the relatively high dispersity of the initial polymers, one cannot exclude formation of intramolecular complexes. The hydrodynamic sizes of the metal complexes are generally below 10 nm in this M/L range. The intermediate zone B in the Figure 10, covers the M/L range from approx. 0.1 to 0.2. At such a metal-to-ligand ratio, we observe polymeric species that have in average at least double or higher molar masses, signifying intermolecular complexation, the supramolecular assembles, however, are still can be considered to be in dissolved state. The average size of the complexes is yet only about 10 to 15 nm. Further increase of the metal-ion content brings us to the zone C that describes the M/L range from ~0.2 to ~0.3, where we basically observe formation of higher order aggregates/assembles with sizes of ~30 to ~50 nm and small amounts of their larger aggregates. The macromolecular assembles consist here of about 50 to 100 initial polymer molecules. The Zone D describes the macromolecular assembles at the M/L ratios close to the critical value (~0.5)—after which no stable solutions/suspensions can be obtained (Zone F). In Zone D, we only observe large macromolecular assembles/aggregates consisting of >>1000 of initial macromolecules with the average sizes >100 nm in diameter. 

### 3.4. Discussion

In the “classical” approaches towards fabrication of metallo-supramolecular systems/assembles, the latter ones are assembled by the coordination interactions form the corresponding ligand-monomeric units. The overall size of the supramolecular assembles will mainly depend on the components concentration, initial M/L ratio, association-dissociation rate etc. While in our case, the assembling mechanism and it’s the dynamics will as well be affected by the polymer nature of the ligand’s carrier. 

It has been shown for classical systems, on the example of ditopic bis-terpyridine ligands coordinated by different metal ions (Fe^2+^, Co^2+^, Ni^2+^), that the overall “chain” size, in terms of amount of monomers in supramolecular assembles will exponentially increase at M/L=1 and be very small at M/L<1, and medium for M/L>1 [44,45]. Furthermore, when either metal ion or ligand is in excess (y≠1) the chain grow will eventually end as soon as one of the components will be consumed. Which is markedly different from the behavior we observe for the polymer-ligand based metallo-supramolecular complexes where the overall working range of M/L ratios is shifted by an order of magnitude towards smaller values with the M/L~0.5 being the critical value after which the system precipitates, while the active chain grow length is observed already at M/L≈0.1. The molar stoichiometry ratio, y=M/L, we have been using along the manuscript, in classical terms, reflects the metal to ligand/monomer molar concentration ratio [44]. Since we initially operate with polymer molecules having tpy ligands in the side chains, we have then recalculated the M/L ratios as number of metal ions per one macromolecule ($M/L^{*}$). By the virtue of numbers, in our case such transformation will result in a simple thousandfold shift of the initially established M/L values. We see then that the active intermolecular complexation, that begins by zone B at M/L≈0.1, correspond to approx.100 metal ions per single macromolecule having in average a 100 ligands groups (100 and 110 for the TerPol and TerFerCop respectively), so in terms of metal to number of ligands calculated based on the actual molar mass, it becomes one to one ratio. The precipitation is then accrued when we have a large excess of free metal ions—approx. 5 metal ions per ligand, or in total ~500 per macromolecule. It was quite surprising to note that even at such small initial polymer concentrations were the polymer solutions can truly be described as dilute ones, we observe such strong intermolecular self-assembly at high salt concentrations (zone C, D). Moreover, as much clearly seen by DLS results, the inter macromolecular complexation begins already at the very small M/L~0.02−0.03 ratios. 

Another interesting aspect, is that, we do not see any specific differences between the complexation behavior of Eu^3+^ or Pd^2+^ metal ions. It is however known, that at the equal ratio of Eu^3+^ ions to tpy sites a 1:1 complex is formed; this complex further contains a number of charge-balancing ions or solvent molecules, as the ancillary ligands (up to a coordination number of 9 in total). In the presence of an excess of ligand moieties, Eu^3+^ is capable of binding with a maximum of three tpy ligands with some dynamic ligand exchange until an equilibrium state is reached (Figure 2a), whereas, Pd^2+^ ions are known to form square-planar complexes in which the metal center should be chelated by the tpy ligand in tridentate fashion (in the case of a 1:1 stoichiometry, one counterion is expected to occupy the fourth position). At a lower Pd^2+^ content the situation might be more complicated and complexes might be formed, in which two tpy ligands are coordinated via two of their *N*-atoms (leaving the third *N*-atom vacant; Figure 2b), so due to the its square-planar geometry and lower coordination number (up to 4), the Pd^2+^ ions should preferably form only unimolecular complexes with the tpy ligands.

Very active intermolecular complexation together with generally equality of the hydrodynamic and macromolecular behavior of the of Eu^3+^ and Pd^2+^ based supramolecular systems could probably be explained from the perspective of polymer-solvent and charge interactions. When substantial amount of metal ions is bound to the ligands distributed along the polymer chain, the solubility (thermodynamic quality of the solvent) of these new polymeric species changes (decreases) tending, thereby, macromolecule to minimize its contact with the solvent molecules causing at certain point a phase separation that should in general provoke formation of compact spherical like particles. A variation of such nanoprecipitation techniques is widely applied in fabrication of polymeric particles/micelles. The “only” difference is that the solvent quality is changes not by addition a non-solvent to the system, but by the addition of metal ions to the polymeric chain. Stable nanoparticle suspensions, from the thermodynamic perspective, can be obtained only in a very narrow metastable region,—the so called “Ouzo” region [46]. It was shown for example that ~100 nm in diameter particles can be obtained from the comparable by molar mass ($M_{w}\approx 40,000\;{\mathrm{gmol}}^{-1}$) poly(methyl methacrylate) based at *c*[*η*] = 0.01 [42]. In our case due to the supramolecular nature of such tpy based supramolecular particles they may further aggregate/assemble either in order to reduce its contact with unfavorable solvent molecules or due to the metal-ligand coordination mechanism. That is well confirmed by the performed microscopy studies, where it is clearly seen that the supramolecular aggregates/assembles consist of individual spherical/globular like particles. Another important aspect to be considered of is associated with the charge effects, that will arise due to the presence of the metal ions/counterions in the polymeric chain and/or solution [6]. At high salt content when most of the binding sites are occupied or unavailable due to steric effects further increase of the salt concentration may lead to the salting out effects [6] which are typical for classical polyelectrolyte systems [47]. Similar behavior was also previously observed for poly(butyl methacrylate)-*co*-2-(1,2,3-triazol-4-yl)pyridine copolymers complexed with Eu^3+^, Fe^2+^ and Co^2+^ metal ions in solution [5,6]. So, at later stages the extensive agglomeration/structuring of the initially formed complexes can be trigged by the solvent quality changes and the charge effects that will accompany the coordination interactions. However, the real contribution each of these effects on the complexation behavior is ought to be discovered.

## 4. Conclusions

Applying mutually reinforcing combination of synthetic and analytical tools, model polymer systems bearing tpy ligands were synthesized and subsequently investigated in detail with respect to their complexation behavior with Eu^3+^ and Pd^2+^ metal ions in dilute solutions. Both metal ions were shown to provoke the intra and the inter macromolecular complexation in solution at principally the same *M*/*L* ratios. It was shown that the supramolecular complexation in addition to the basic parameters that define it from the classical coordination chemistry will also be influenced by the polymer nature of the ligand’s carrier and associated with the polymer solvent and charge interactions in solution.

## Data Availability

Not applicable.

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
