# Peer review of "Metallo-Supramolecular Complexation Behavior of Terpyridine- and Ferrocene-Based Polymers in Solution—A Molecular Hydrodynamics Perspective"

_polymers, 2022, doi:10.3390/polym14050944_

Round 1

Reviewer 1 Report

- The authors should be careful in the figure caption numbers (fig.  4 and 5) on pages 10 and 11(are the same as the figure number on pages 8 and 9). 

- Section SI should cover a brief description of each figure.

 - Section Results and discussions should be split into results/discussions. The title "results" is not present.

Reviewer 2 Report

This manuscript describes metallo-supramolecular complexation behavior of terpyridine- and Ferrocene-based polymers in dilute solutions. Combination of sedimentation-diffusion analysis in the analytical ultracentrifuge, dynamic light scattering, viscosity and density measurements, morphology studies by electron microscopy enables mapping the conformational states/shapes, the corresponding hydrodynamic and macromolecular characteristics of metallo-supramolecular assemblies at various metal/ligand ratios. Authors suggest the complexation starts at the ratio of 1/10 (M/L) and the precipitation occurs at the ratio higher than 1/2 (M/L). Authors perspective the intramolecular complexation is caused by the polymer-solvent and charge interactions.

              Introduction on metallopolymers is well written in compact. Experimental section is also written for details. The determination of the molar ration of M/L in TerFerCop by NMR will be correct. The different molecular weights of TerPol and TerFerCop determined by SEC analysis and sedimental-diffusion analysis are also well explained.

              My question is why authors chose Eu3+ and Pd2+ as the metals. As the ligand is terpyridine, transition metals with an octahedral coordination geometry should be selected to obtain the 1:2 complex of metal and terpyridine. Eu3+ can be complexed with 8-10 coordination atoms and especially strongly coordinated with oxygen atoms. Pd2+ also has four coordination geometry, not six coordination. To help the understanding of readers, authors should illustrate the complexation structures authors anticipate.

              As this manuscript includes useful information on the metallo-supramolecular complexation behavior, I suppose it should be accepted for publication, but authors should add some figures to explain their proposed assembling mechanisms.

Minor errors:

  • The abbreviations of polymers are not unified at all in the manuscript.

TerFerPol, TerFerPoly, or TerFerCop?

TerPol or TerPoly?

  • In Figure 3 and the corresponding explanation, “terpyridine homopolymer” and “terpyridine-ferrocene copolymer” are used, but authors should use the abbreviations.
  • In page 7, correct (0.0.1 < M/L < 0.1) to (0.01 < M/L < 0.1).
  • In page 14, correct “will mainly be depend on” to “will mainly depend on”.
  • In the caption of Figure 7, the explanation of A and B is missing.

Reviewer 3 Report

This article report the self-assembly behavior of poly(methacrylate)-based copolymers containing ferrocene and/or terpyridine units upon interactions with Eu(III) and Pd(II) ions in dilute solutions. The study is mainly based on of sedimentation-diffusion analysis to access the effect of the interaction with Eu(III) or Pd(II) ions on the size and morphology of the polymers. Compared to terpyridine supramolecular polymer based on bis terpy monomers that polymerize at a M/L ratio ~ 1, the reported copolymers present aggregation at much lower M/L ratio ~ 0.1 and precipitate at ?/? ≈ 0.5. This drastic difference in behavior is interpreted by the high density of terpyridine ligand on each polymeric chain that enable coordination of multiple terpy and intermolecular self-assembly. The study is well conducted and provides interesting insight on the effect of terpyridine coordination on the polymer. I recommend for publication if the following point are addressed: 

1) The nature of the complexes formed by addition of the cations is not clearly defined. While Eu(III) due to its coordination number of 9 can form bis or tris terpyridine complexes intramolecularly or even intermolecularly, Pd(II) due to its square planar geometry can only generate intramolecular mono terpy complexes. It is quite surprising that both cation seems to generate rather similar aggregation behavior. The authors should evaluate the nature of the complexes formed by addition of the cations and comment the M/L ratio in accordance to the stoichiometry of the complexes for each cation. 

2) Integration of the 1H NMR spectra should be provided in Fig 2 or in the SI. 

3) Typo error in p3. “1HNM “should be “1H NMR”. 
